# Poly-*α*, *β*-*d*, *l*-Aspartyl-Arg-Gly-Asp-Ser-Based Urokinase Nanoparticles for Thrombolysis Therapy

**DOI:** 10.3390/molecules28062578

**Published:** 2023-03-12

**Authors:** Shuangling Chen, Meng Liang, Chengli Wu, Xiaoyi Zhang, Yuji Wang, Ming Zhao

**Affiliations:** School of Pharmaceutical Sciences, Capital Medical University, Beijing 100069, China

**Keywords:** urokinase, thrombolytic, targeted, nano delivery system, RGDS

## Abstract

The most concerning adverse effects of thrombolytic agents are major bleeding and intracranial hemorrhage due to their short half-life, low fibrin specificity, and high dosage. To alleviate bleeding side effects during thrombolytic therapy which would bring about the risk of aggravation, we try to find a novel biodegradable delivery nanosystem to carry drugs to target the thrombus, reduce the dosage of the drug, and system side effects. A novel urokinase/poly-*α*, *β*-*d*, *l*-aspartyl-Arg-Gly-Asp-Ser complex (UK/PD-RGDS) was synthesized and simply prepared. Its thrombolytic potency was assayed by the bubble-rising method and in vitro thrombolytic activity by the thrombus clot lysis assay separately. The in vivo thrombolytic activity and bleeding complication were evaluated by a rat model of carotid arteriovenous bypass thrombolysis. The thrombolytic potency (1288.19 ± 155.20 U/mg) of the UK/PD-RGDS complex nano-globule (18–130 nm) was 1.3 times that of commercial UK (966.77 ± 148.08 U/mg). In vivo, the UK/PD-RGDS complex (2000 IU/kg) could reduce the dose of UK by 90% while achieving the equivalent thrombolysis effect as the free UK (20,000 IU/kg). Additionally, the UK/PD-RGDS complex decreased the tail bleeding time compared with UK. The organ distribution of the FITC-UK/PD-RGDS complex was explored in the rat model. The UK/PD-RGDS complex could provide a promising platform to enhance thrombolytic efficacy significantly and reduce the major bleeding degree.

## 1. Introduction

Thrombotic diseases seriously endanger the health and life of patients. It is believed that the best way to improve the survival rate and reduce the mortality rate of patients is immediate, early detection, and effective thrombolytic therapy [1]. In recent years a variety of technical methods have emerged, including drug thrombolytic therapy, interventional thrombolytic therapy, and interventional thrombolysis. However, thrombolytic therapy is still an important, irreplaceable, and fundamental measure for the treatment of thromboembolism [2]. Thrombolytic agents in clinical use are Plasminogen activators (PAs) to treat thromboembolism, including streptokinase (SK), urokinase (UK), alteplase (RT-PA), tissue plasminogen activator (tPA) and tenepase (TNK-TPA), etc. UK is an effective thrombolytic drug that has a low price and is widely used in primary hospitals [3]. However, due to the short half-life (2–20 min) [4], PAs need more doses to be administered. More doses of PAs could cause more serious system hemorrhage because the low thrombus-specificity of Pas activates both fibrin-bound and circulating plasminogen creating a serious risk of hemorrhage. These factors could cause a side effect risk of harmful bleeding complications and lead to the aggravation of the disease. Bleeding complications of Pas bring difficulties and risks to the clinical application of Pas including r-tPA with fibrin specificity [5,6]. On the whole, decreasing the bleeding complications of PAs could be very significant and is urgently needed for the clinical application of PAs.

Targeted delivery systems of PAs could target PAs to the site of the thrombus and at the same time reduce the dose of PAs. So, the systematic generation of broad matrix-specific fibrinolytic enzymes associated with bleeding complications could decrease and targeted delivery systems of PAs could be a better way to avoid or resolve dose-induced side effects [4]. Many targeted delivery systems have been performed to target PAs to the site of the thrombus, release PAs and perform effective thrombolytic treatment. Targeted delivery systems were generally prepared by directly bonding the thrombus-targeted ligands to PAs or the surface of drug carriers, such as liposomes [7,8], magnetic targeting delivery systems [9,10], microbubbles [11,12,13], polymer nanoparticles [2,9,14,15,16] and new inorganic-organic hybrid nanoparticles for deeper or continuous release in recent years [9,10]. The polymers in local delivery systems of PAs include PLGA Polymers, polyglutamic acid peptide dendrimer, chitosan derivatives, PEG, etc. [9,16,17,18]. These thrombus-targeted ligands include fibrin-specific anti-fibrin antibodies [19,20], vMF factor-specific alkaline gelatin [21], p-selectin-specific fucoidan [2,4], activated platelets-specific RGD sequence peptide, etc. [8,9,12,13,16]. Then, by wrapping or bonding PAs, these targeted delivery systems were constructed into thrombolytic targeting delivery systems. In addition, thrombin-sensitive peptides and pH-sensitive phenyl imine bonds also were used to develop PAs delivery systems and could be ruptured at the thrombus of the stroke to perform targeted thrombolytic therapy [3,4,18,21].

RGD sequence peptides exist in the α-chain of the fibrinogen and could specifically recognize Glycoprotein (GP) IIb/IIIa receptor on the activated platelet membrane. RGD sequence peptides have drawn much attention from researchers in the diagnosis of thrombosis and targeted thrombolytic therapy [22]. The cyclic RGD (cRGD) functionalized liposome has been used to carry urokinase to target thrombus in vivo thrombolysis study [8]. However, intravascular liposomes are limited due to stability problems. The major challenges in the development of liposomal PAs-targeting delivery systems are cost and still ineffective treatment [7,8]. In other research, it was shown that thrombolytic therapy by targeted microbubbles containing RGD sequence peptides under ultrasound could destroy the fibrillary network structure of the thrombus [13] and enhance the dissolution of the thrombus, while targeted nano-bubbles have a higher thrombolytic rate and penetrate deeper into thrombus than targeted micron bubbles [12]. Additionally, polymer nanoparticles are relatively stable carriers. Targeted nanoparticles, such as mesoporous carbon nanomaterials [10], poly(lactic-co-glycolic acid) magnetic nanoparticles [6], and chitosan nanoparticles [16]. Furthermore, non-bonding complex PAs-targeting delivery systems, such as the PAs complex [23,24], were used to target thrombus.

In our previous work, based on the specificity of RGD sequence peptides on activated platelets, multiple RGDS molecules have been bonded to the highly biodegradable poly-*α*, *β*-*d*, *l*-aspartic acid (PD) [25,26,27]. PD-RGDS with a high grafting rate of 46% was prepared to have a specific affinity for activating platelets [26]. In this study, we based on the interaction between proteins and constructed a UK/PD-RGDS complex delivery system (Figure 1). We characterized the UK/PD-RGDS complex delivery system by Zeta Sizer and TEM. The thrombolytic potency of the UK/PD-RGDS complex was measured by the bubble-rising method. The in vivo thrombolytic activity, the bleeding complications, and organ distribution of UK/PD-RGDS were evaluated to investigate the thrombolysis efficacy and the side effects via male Wistar rats. Our study provided a foundation for the development of novel delivery systems for thrombolytic therapy.

## 2. Results

### 2.1. Preparation of UK/PD-RGDS Nanosystem

PD-RGDS was synthesized by the condensation reaction of PD and HCl·Arg(Tos)-Gly-Asp(OBzl)-Ser(Bzl)-OBzl and the removal of protective groups at low temperatures. Its structure was characterized by ^1^H NMR spectra, IR, and amino acid analysis. Its purity was determined by HPLC (7 mg/mL, Ultrahydrogel 120 columns, 7.8 × 30, 35) with a refractive index detector, eluted with 0.1 N NaNO_3_ with a flow rate of 0.5 mL/min. The retention time of PD-RGDS was 11.5 min. The results of the amino acid analysis gave Asp:Arg:Gly:Ser = 3.2:1.0:1.2:1.2. It showed that 46 aspartic acids among 100 aspartic acid units were connected to RGDS.

The UK/PD-RGDS complex was simply prepared by mixing UK and PD-RGDS at 4 °C for 1 h. UK/PD-RGDS complexes with 1:5, 1:3, and 1:1 (*w*/*w*) in 10 mM PBS buffer (pH 7.4) were obtained to explore the effect of the complexation ratio on the system of the UK/PD-RGDS complex. These UK/PD-RGDS complexes were opalescent, uniform, and stable. Their z-average sizes, the size distribution of particles, and Zeta potential were measured and shown in Table 1. On the whole, the z-average sizes of all UK/PD-RGDS complexes (270–277 nm) were smaller than that of PD-RGDS (278 ± 3.772 nm). Their size distribution was narrow after the complexation of UK and PD-RGDS. The smaller sizes and narrow size distribution of the UK/PD-RGDS complexes implied that UK and PD-RGDS have a close interaction. Besides, the zeta potential of the UK/PD-RGDS complex (−15.2 ± 0.830 mV) was between that of the UK (−8.80 ± 0.285 mV) and PD-RGDS (−17.1 ± 0.993 mV) and closer to the zeta potential of PD-RGDS. The zeta potential suggested that both PD-RGDS and UK exposed on the surface of UK/PD-RGDS complexes and the ratio of PD-RGDS exposed on the surface should be more than UK.

The z-average size, size distribution, and Zeta potential of UK/PD-RGDS complexes in proportions of UK/PD-RGDS (1:5, 1:3, 1:1 *w*/*w*) were measured for 48 h to explore the different component ratios on the stability of complexes as shown in Figure 1A–C. The z-average sizes therein showed that the UK/PD-RGDS complex (1:1 *w*/*w*) (272.0 ± 4.66 nm) has a smaller range of variation over 48 h compared to the other two UK/PD-RGDS complexes ratio. So is the distribution and Zeta potential. Thus, we think that the UK/PD-RGDS complex (1:1 *w*/*w*) was the most stable according to the z-average size, size distribution, and Zeta potential. Transmission electron microscopy (TEM) revealed that the UK/PD-RGDS complex (1:1 *w*/*w*) appeared in the solid near circular spheres (18–131 nm) with a narrow size distribution, as shown in Figure 1D. Then, the complex of UK and PD-RGDS (1:1 *w*/*w*) was selected for the following experiments.

### 2.2. Thrombolytic Potency of UK/PD-RGDS Complex

The bubble-rising method was adopted to determine the thrombolytic potency of UK after complexation according to the Pharmacopoeia of the People’s Republic of China (2015).

The fibrin clot formed under the action of thrombin and the time of the small bubbles rising to half the volume of the reaction system from the bottom of the fibrin clot is related to the concentration of UK in Figure 2A. The logarithm of urokinase concentration showed a linear relationship with the logarithm of reaction time during the concentration of 3.53–14.12 U/mL (y = −0.2184x + 2.8094, r^2^ = 0.9930). The thrombolytic potency results showed that the thrombolytic potency of the UK/PD-RGDS complex was 1288.19 ± 155.20 U/mg. Its thrombolytic potency increased and was 1.33 times of UK (966.77 ± 148.08 U/mg) at the same dose, which could not be explained.

To further confirm the increasing thrombolytic potency of the UK/PD-RGDS complex, it was evaluated again by the Agarose-fibrin plate method. In the Agarose-fibrin plate method, agarose was used as a support and a fibrin plate was formed under the action of thrombin. Fibrin was decomposed by urokinase to generate a transparent circle at a certain temperature in Figure 2B. The reaction temperature was optimized by comparing at room temperature for 18 h, room temperature for 24 h, 37 °C for 3 h, and 37 °C for 24 h. The size of the transparent circle obtained at room temperature for 24 h is moderate, not easy to overlap and the error is small. Thus, room temperature for 24 h was selected in the following tests.

The areas of the transparent circles showed a linear relationship with the logarithm of UK concentration during the concentration of 200–800 U/mL (y = 175.07x − 270.4, r^2^ = 0.9971). The thrombolytic potency of UK/PD-RGDS was 1290.80 ± 85.78 U/mg and 1.28 times of UK (1005.48 ± 66.68 U/mg) by the Agarose-fibrin plate method. Due to the thrombolytic potency results by two methods, the thrombolytic potency of UK after loading on the nanosystem of PD-RGDS increased and was consistent with each other.

Thus, the increasing thrombolytic potency of the UK/PD-RGDS complex was confirmed.

### 2.3. In Vivo Thrombolytic Activity of UK/PD-RGDS Complex

The in vivo thrombolytic activity of the UK/PD-RGDS complex by injection administration was evaluated in a rat model of carotid arteriovenous bypass thrombolysis (Figure 3A) [28]. The thrombus was prepared first (Figure 3B) and put into the carotid arteriovenous bypass by assembling. Via operation, the in vivo rat model of carotid arteriovenous bypass thrombolysis was established. After UK or the UK/PD-RGDS complex was injected, the blood began to circulate via carotid arteriovenous bypass. The weight of the thrombus in the bypass would decrease under the thrombolytic action of UK or the UK/PD-RGDS complex. Then the reduction in the thrombus weights after 60 min was used to evaluate the thrombolytic activity. In the evaluation, the rats were divided into three groups. NS (3 mL/kg) was used as the blank control group. The commercial UK (20,000 IU/kg) at a clinical dose (20,000 IU/Kg) was used as the positive control and the UK/PD-RGDS complex (2000 IU/kg) was used as the test group.

Figure 3C showed the in vivo thrombolytic activity results of both the positive control and our sample. Initially, we used a dosage of 20,000 IU/Kg (the clinical dose of UK) in the in vivo assay to compare the thrombosis activity between our sample (the UK/PD-RGDS complex at 1:1 ratio) and the commercial UK. However, the result showed that both of positive control (UK) and our sample (UK/PD-RGDS complex) are active and have similar in vivo thrombolytic potency (Figure 3C). We thought that both UK and our sample may reach their maximal thrombolytic activity at 20,000 IU/kg. To assess whether our sample is more potent than commercial UK, we lowered the dose of our sample to 2000 IU/kg and evaluated them in a second in vivo assay. Compared with the NS group, there was no significant difference in thrombus weight reduction in the UK (2000 IU/kg) group (*p* > 0.05). The results showed that a low dose of our sample was as effective as high doses, whereas a low dose of UK was inactive (Figure 3D). Therefore, we claimed that the thrombolytic activity of the UK/PD-RGDS complex was enhanced compared to UK.

### 2.4. Tail Bleeding Time of UK/PD-RGDS Complex

In the evaluation of the tail bleeding time of the UK/PD-RGDS complex, the rats were divided into three groups. The tail bleeding time of rats before administration was used as a blank control group. The UK group (20,000 IU/kg) was used as a reference control. The UK/PD-RGDS complex (2000 IU/kg) group was used as the test group. The tail bleeding time results (Figure 4) showed that the tail bleeding time of the UK/PD-RGDS complex group (428 ± 137 s) was significantly lower than in the UK group (692 ± 141 s) (*p* < 0.05, n = 5). The UK/PD-RGDS complex could reduce the side effect of bleeding.

### 2.5. In Vitro Thrombus Clot Lysis Assay

Thrombus clots were first prepared from rat arterial blood (65.0–75.0 mg). Thrombus clots were divided into three groups, the NS group, the group of (100 IU/mL UK), and the group of the UK/PD-RGDS complex (100 IU/mL) at 37 °C for 3 h. The reduced weights of thrombus clots present the lysis activity to thrombus clots of these compounds. Figure 5 showed the in vitro “thrombolytic activity” results of UK and the UK/PD-RGDS complex at a lower dose (n = 9). Not surprisingly, both groups showed similar potency in this assay, since they contained the same dose of functional agent UK. Our results indicated that PD-RGDS did not directly enhance the activity of UK by forming nanoparticles with UK. The UK/PD-RGDS complex is designed to improve the pharmacokinetic profile of commercial UK by increasing its stability and delivering it to the thrombus.

### 2.6. Organ Distribution Study

FITC-UK (F/P = 4.73) was performed in the organ distribution study. In the organ distribution study, the rats were divided into three groups. The NS group was used as a blank control. FITC-UK (8000 U/kg) was used as a reference control. FITC-UK/PD-RGDS complex (8000 U/kg) was used as the test group. Results in Figure 6 showed that FITC-UK in both FITC-UK or FITC-UK/PD-RGDS complex was mainly distributed in the liver and blood, while ratios of FITC-UK were very low in the spleen and lungs. The FITC-UK ratio of the FITC-UK/PD-RGDS complex group in the thrombus was significantly improved (*p* < 0.05, n = 5) compared with the FITC-UK group and was 2.3 times than the FITC-UK group. It can be seen that FITC-UK in the complex group was significantly accumulated at the site of the thrombus compared with the FITC-UK group, which suggested the thrombus targeting effect of the FITC-UK/PD-RGDS complex. It was because the RGDS in the FITC-UK/PD-RGDS complex highly binds to the activated platelet membrane GPIIb/IIIa in thrombus [26]. The ratio of the FITC-UK/PD-RGDS complex in the blood was also significantly increased (*p* < 0.01, n = 5) and 1.6 times of FITC-UK.

## 3. Discussion

Thrombolytic therapy is still the most basic treatment method and the fundamental measure to treat thrombosis. Plasminogen activator is currently an effective thrombolytic drug in clinical practice. Plasminogen activators (PAs) systematically activate plasminogen to become plasmin, then the produced plasmins degrade fibrinogen and fibrin in the clots, and decompose the thrombus. When excessive plasmins are produced, bleeding of different degrees occurs. Because PAs was a group of proteases and would be degraded quickly in the blood, PAs have short half-life periods (2–20 min). To achieve the effect of thrombolytic therapy, large doses of drugs were used to reach the high blood concentration for treatment, which increases the risks of bleeding side effects and non-specific toxicity. Reducing bleeding side effects could improve the safety of the medication and reduce pressure for doctors and the risk of bleeding for patients, which was urgent and meaningful.

A targeted delivery system could target thrombolytic drugs to the thrombus site and release thrombolytic drugs for effective thrombolysis. So, targeted delivery systems could reduce the dose to achieve the treatment effect of thrombolytic therapy. Meanwhile, a targeted delivery system will also reduce the bleeding side effects caused by the increasing dose of PAs, and is the best way to avoid or solve the side effects caused by the dose [1].

The RGDS sequence peptide exists at 572–575 on the alpha chain of fibrinogen. It can specifically recognize the glycoprotein (GP) IIb/IIIa receptor on activated platelet membranes, which has been widely studied for the diagnosis of thrombosis and targeted thrombolytic therapy in recent years. RGDS can competitively bind activated platelets, thus preventing the fibrin bridging, and has the effect of inhibiting platelet aggregation. Here, PD-RGDS was a safe and biodegradable polyamino acid carrier; the main chain is biodegradable poly-*α*, *β*-*d*, *l*-aspartic acid, and the side chain consists of amino-group of poly-*α*, *β*-*d*, *l*-aspartic acid bonded with the carboxyl group of RGDS. So PD-RGDS also could specifically recognize the glycoprotein (GP) IIb/IIIa receptor on the activated platelet membrane. Moreover, in this study, the PD-RGDS carried plenty of RGDS motifs (grafting ratio 46%). The Effect of PD-RGDS on GPIIb/IIIa expression results also confirmed that PD-RGDS (10^−5^ M) could reduce to one-thousandth of the concentration of RGDS while achieving the equivalent binding to the GPIIb/IIIa receptors on the platelet surface as RGDS (2.5 × 10^−2^ M) [26]. Because plenty of RGDS motifs are endowed with high specific binding to activated platelets, PD-RGDS could target thrombus better. In addition, the transmission electron microscopy results showed that PD-RGDS existed as nanoparticles of 60–108 nm which was more beneficial to construct a nano-sized drug delivery system than a micron-sized carrier.

The preparation method of the UK/PD-RGDS complex was simple, green, and has a short mixing time, which better protected the activity of UK than the UK conjugation delivery system. The z-coverage sizes and zeta potentials results implied that the UK/PD-RGDS complex could exist stably for 3 days. Moreover, in UK/PD-RGDS complexes UK and PD-RGDS had a close interaction because the UK/PD-RGDS complex has smaller sizes and narrow size distribution than UK and PD-RGDS. The zeta potential results showed both UK and PD-RGDS are exposed on the surface of the UK/PD-RGDS complex. We think the complexation and closer binding of UK and PD-RGDS could cause the changing of UK and PD-RGDS conformation and be in favor of the increase in UK potency. This may be the reason for the increasing thrombolytic potency of the UK/PD-RGDS complex. In addition, the transmission electron microscopy results showed that UK/PD-RGDS existed as nanoparticles of 18–131 nm. Fibrin clots highly inhibit the penetration of particles of 1 μm or larger into fibrin clots [29], which has allowed the nanosized system to accelerate thrombolytic therapy without causing microbubbles and holes. The nanosized UK/PD-RGDS was more beneficial to penetrate the thrombus for thrombolysis than the micron-sized carrier.

Platelets are the main targets of thrombus [7]. RGD sequence peptides could specifically bind to activated platelets by targeting GPIIb/IIIa on the surface of platelets [30,31]. PD-RGDS containing 46% RGDS has specifically adhered to activated platelets [26]. Then PD-RGDS containing these RGDS motifs loaded UK specifically to the thrombus site and then UK was concentrated to dissolve the local thrombus. Therefore, less dose of the UK/PD-RGDS complex (2000 IU/kg) could show significant thrombolytic activity as the free urokinase at the dose of 20,000 IU/kg in a rat model of carotid arteriovenous bypass thrombolysis. It could be explained by the increasing thrombolytic potency, high ratio targeting factors, and nano-sized particles. Firstly, the thrombolytic potency of the UK/PD-RGDS complex increased compared with the UK by the Bubble-rising method. That is to say, the structure of the UK/PD-RGDS complex could help to increase the thrombolytic potency of the UK and cause the stronger thrombolytic activity of the UK/PD-RGDS complex group in vivo to a certain extent. Moreover, the grafting rate of RGDS in PD-RGDS is very high and reaches up to 46%. The high grafting rate of RGDS in PD-RGDS could help UK better to concentrate on the thrombus and increase the thrombolytic activity. After complexation with PD-RGDS, UK at the dose of 2000 IU/kg had significant thrombolytic activity. However, the in vivo UK group (2000 IU/kg) has no significant thrombolytic activity compared with the NS group. It suggested that the thrombolytic activity of the UK/PD-RGDS complex group (2000 IU/kg) was 10 times of the UK group. In addition, the nanoscale UK/PD-RGDS complex system could be beneficial to dissolve thrombus. It showed that polymer conjugation by grafting with targeted motifs is a good method as the drug carrier when we construct a targeted nano-delivery system.

The tail bleeding time results indicate that the UK/PD-RGDS complex could reduce the side effect of bleeding. Meanwhile, the UK/PD-RGDS complex could improve thrombolytic activity. Therefore, this study achieved the purpose of our expected research design. The tail bleeding times after NS administration was significantly higher than that before NS administration (*p* < 0.05, n = 5) (Figure 4), indicating that the tail bleeding time of the blank control group was increased. It was perhaps caused by heparin (140 U/kg) added to the carotid arteriovenous bypass of rats.

The interaction between UK and PDRGDS could not be analyzed because the determination of protein interactions by isothermal calorimetric titration requires the unavailable UK sample with a single molecular weight; our UK is a mixture of high molecular weight UK (Mw 54,000) and low molecular weight UK (Mw 33,000).

## 4. Materials and Methods

### 4.1. Materials

All amino acids were purchased from Sichuan Sangao Biochemical Co., Ltd. (Chengdu, China). Urokinase for injection (100,000 units) was purchased from Peking University Gaoke Huatai Pharmaceutical. Bovine fibrinogen Standard (Lot 140607-201841), Bovine thrombin Standard (Lot 140605-201526), Bovine fibrinogen Standard (Lot 140606-201826), bovine fibrinogen Standard (Lot 140606-201826), Bovine fibrinogen standard (Lot 140606-201826), Bovine fibrinogen standard (Lot 140606-201826), Bovine fibrinogen standard (Lot 140606-201826) and Urokinase standard (Batch No. 140604-201224) were purchased from China National Institute for Food and Drug. Bovine thrombin(SLBV3604) and Fluorescent isothiocyanate yellow (FITC)were purchased from Sigma Company (Shanghai, China). Agarose (Batch No. 424G056) was acquired from Beijing Solebo Technology Co., Ltd. (Beijing, China). Barbiturate-sodium chloride buffer (pH 7.8, DZ331) was purchased from Xi’an Hutt Biological Company (Xian, China). Trimethylol aminomethane buffer (pH 9.0) was purchased from Beijing Regen Biotechnology Co., Ltd. (Beijing, China). Other reagents were purchased from Sinopharm Chemical Reagent Co. Ltd. (Shanghai, China).

### 4.2. Preparation of Poly-α, β-d, l-Aspartyl-Arg-Gly-Asp-Ser (PD-RGDS)

The preparation of poly-*α*, *β*-*d*, *l*-aspartyl-Arg-Gly-Asp-Ser was carried out according to the method in the literature [26]. In short, 82.8 mg of PD was dissolved in 1 mL of anhydrous DMF, then 97 mg of HoBt and 360 mg of HCl·EDC were added to an ice bath. After 0.5 h, HCl·Arg(Tos)-Gly-Asp(OBzl)-Ser(Bzl)-OBzl was added and the pH value was adjusted to 9. After 24 h, the reaction solution was dried, extracted by ether, and washed separately with 5% KHSO_4_, and water three times. The solid was dried at 37 °C under reduced pressure for 48 h and provided 197 mg of the yellowish powder. The yellowish powder was dissolved and mixed in 8 mL of CF_3_CO_2_H:CF_3_SO_3_H (3:1) at 0 °C for 75–90 min. Then it was triturated with 150 mL of ether and the residue was mixed with water and dissolved until the pH value was adjusted to 7. After centrifugation for 30 min, the supernatant was dialyzed for 3 days with ultrapure water and lyophilized to provide 197 mg (72%) of the title compound as a white powder.

### 4.3. Preparation of UK/PD-RGDS Complex

An amount of 8 mg of PD-RGDS and 8 mg of UK were mixed in 3 mL of pH 7.4 PBS buffer (10 mM) and stirred at 4 °C for 1 h to prepare the solution of 6.667 IU/mL UK/PD-RGDS complex.

### 4.4. Morphology of UK/PD-RGDS Complex

The sizes and Zeta potentials of samples (2.67 mg/mL)in pH 7.4 PBS buffer (10 mM) were determined in the automatic measurement mode on Malvern’s Zeta Sizer (Nano-ZS90). The sizes and morphology of UK/PD-RGDS complex particles were observed by transmission electron microscopy (JEM-2100, Japan). The solutions of the UK/PD-RGDS complex (10^3^, 10^2^, 10^−2^, 10^−5^, 10^−7^, 10^−9^ nM) were prepared and dropped onto a formvar-coated copper grid as TEM samples. Then a drop of ethanol was added. The copper grid is first placed in the air to dry completely. The samples were observed by transmission electron microscopy (JSM-6360 LV, JEOL, Tokyo, Japan) with an electron beam acceleration voltage of 120 kV. All samples were prepared in three copies.

### 4.5. Bioassays of UK/PD-RGDS Nanosystem

#### 4.5.1. Bubble-Rising Method

The test of the Bubble-rising method was performed according to Pharmacopoeia of the People’s Republic of China, Part II (2015 Edition); 6.67 mg/mL bovine fibrinogen standard in barbiturate-sodium chloride buffer (pH 7.8), 6.0 bp/mL bovine thrombin standard in barbiturate-sodium chloride buffer (pH 7.8), and 1 casein unit/mL bovine plasminogen in Tris (Hydroxymethyl) aminomethane buffer solution (pH 9.0) was prepared first. Bovine thrombin and bovine plasminogen were mixed with equal volume and the mixed solution was obtained. A 60 units/mL standard solution of urokinase in a barbiturate-sodium chloride buffer (pH 7.8) was also prepared.

The UK/PD-RGDS complex prepared by method 2.2 was quantitatively diluted with barbiturate-sodium chloride buffer (pH 7.8) to the concentration of 60 units/mL UK. The sample of UK control was also prepared according to method 2.2; 0.3 mL of bovine fibrinogen was added to each tube and put at 37 ± 0.5 °C in a water bath. Then 0.9 mL, 0.8 mL, 0.7 mL, and 0.6 mL barbiturate-sodium chloride buffer (pH 7.8) were added, respectively; 0.1 mL, 0.2 mL, 0.3 mL, and 0.4 mL of UK standard solution were added successively. After that 0.4 mL of the mixed solution was added, and each tube fully oscillated until the reaction system was full of bubbles and the bubbles stay in the system. The reaction system usually condenses in 30~40 s. The end of timing was recorded when the small bubbles rose to half the volume of the reaction system in the clot. All the samples were carried out in triplicate. The potency of the UK samples and UK/PD-RGDS complex samples were measured and determined by converting measurements to the thrombolytic potency through a standard curve between the logarithm of time versus the logarithm of the concentration of urokinase.

#### 4.5.2. Agarose-Fibrin Plate Method

A concentration of 8 mg/mL bovine fibrinogen standard, 1 mg/mL bovine thrombin standard, and 8 mg/mL agarose standard in 10 mM PBS buffer (pH 7.4) were prepared separately. Then 8 mg/mL agarose was heated in a microwave oven until boiled. After the agarose was completely dissolved, the agarose was placed in a hot water bath at 52 °C for later use; 800 U/mL, 600 U/mL, 400 U/mL, and 200 U/mL UK standard in 10 mM PBS buffer solution (pH 7.4) were prepared separately. UK samples and UK/PD-RGDS samples were diluted to an appropriate concentration with 10 mM PBS buffer (pH 7.4). Three Petri dishes (9 cm in diameter) were taken and numbered; 18 mL of agarose solution was added to a small beaker. Then 1 mL of bovine thrombin and 1 mL of bovine fibrinogen were added to the beaker. They were shaken thoroughly and quickly poured into a disposable Petri dish. A homemade punch was used and the mixtures stayed at room temperature for 1 h. After the agarose was completely solidified, the liquid in the well was drained; 5 μL of 800 U/mL, 600 U/mL, 400 U/mL, and 200 U/mL UK standard was added separately into the holes. After the mixture has been placed at room temperature for 24 h, transparent rings on the agarose-fibrin plate were observed. The potency of UK samples and UK/PD-RGDS complex samples were measured and determined by converting measurements to the thrombolytic potency through a standard curve between concentrations of UK versus the area of the transparent rings.

### 4.6. In Vitro Thrombus Clot Lysis Assay

The in vitro thrombus, clot lysis assay was carried out according to the procedure reported earlier [28]. Male Wistar rats (220 g ± 10 g) were anesthetized with pentobarbital sodium (20%, 7 mL·kg^−1^, i.p.). The right carotid artery was isolated and the whole blood was collected in centrifuge tubes. The whole blood was injected into a flexible rubber hose (D 1.7 cm) containing a helix (L 15 mm; D 1.0 mm). After 40 min the thrombus with helix was carefully removed and suspended in the tri-distilled water for 1 h at room temperature. The surface water was removed using filter paper and the thrombus was weighed precisely. Then they were immersed into 8 mL NS, UK (100 IU/mL), or UK/PD-RGDS complex (100 IU/mL), respectively, at 37 °C at 70 rpm in a shaker for 3 h. The thrombi were removed and the surface water was gently removed by filter paper. The reduced weight of the thrombus was used to compare the degree of thrombus clot lysis.

### 4.7. In Vivo Thrombolytic Activity

Male Wistar rats (210–250 g) were anesthetized with pentobarbital sodium (20%, 7 mL·kg^−1^, i.p.). The right common carotid artery and the left vein were operated on and isolated. The whole blood was collected from the right common carotid artery and used to prepare the thrombus clots for 40 min. The surface blood of the thrombus with helix was removed using filter paper and the thrombus was weighed precisely. The thrombus was put into a polyethylene tube as an external circulation pipeline between the right common carotid artery and the left vein. These pipelines were filled with heparin sodium (50 IU/mL NS solution). One end was inserted into the left internal jugular vein and after the heparin sodium (200 U/kg) was injected the other end was inserted into the right carotid artery. NS (3 mL/kg), UK (20,000 IU/kg), or UK/PD-RGDS complex (2000 IU/kg) were injected near the venous end. After the blood was circulated for 60 min, the thrombus was taken out and weighed after the surface blood was absorbed. The reduced weight of the thrombus was used to represent their thrombolytic activity in vivo.

### 4.8. Determination of the Tail Bleeding Time

The tail bleeding time was assayed as described previously with a few small modifications [32]. The operating method was referred to in 4.7. The difference is that the dose of heparin sodium (200 U/kg) was adjusted to 140 U/kg and only used to fill the tube instead of intravenous injection. Bleeding times were measured at 40 min before and after the thrombolytic treatment. The rat tail was cut off at 1 mm near the tail tip and placed in 25 mL of normal saline at 37 °C. The occurrence of uniform and continuous bloodlines was taken as the beginning of timing. The complete cessation of bleeding was recorded as the bleeding time. If the bleeding does not stop at 1800 s, it is classified as 1800 s.

### 4.9. Organ Distribution Study

Firstly, UK was labeled by FITC as in previous articles [33]; 10 mg of urokinase was dissolved in 1 mL of 10 mmol/L PBS (pH 7.1); 2.6 mg of fluorescence isothiocyanate yellow (FITC) was dissolved in 100 μL of 0.5 mol/L carbonate buffer (pH 9.5). FITC was dropped into urokinase and stirred in a shading environment at room temperature for 4 h. Then the mixture was centrifuged at 2500 r/min for 25 min. The supernatant was dialyzed in 10 mmol/L PBS buffer (pH 8.0) for 2–4 h. After dialyzation, FITC-UK was purified by Sephadex G50 column and eluted with 10 mmol/L PBS buffer (pH 7.1). The fluorescence of FITC-UK was measured by F-2500 Fluorescence Spectrophotometer. FITC-UK was diluted appropriately until its OD280 was close to 1.0. Its OD value was read at 495 nm and 280 nm. F/P value was calculated according to the following formula: F/P = 2.87 × OD_495_/(OD_280_ − 0.35 × OD_495_).

In vivo organ distribution was evaluated as in previous articles [34]. Male Wistar rats (250–300 g) were anesthetized with pentobarbital sodium (20%, 7 mL·kg^−1^, i.p.). After the right common carotid artery was separated, filter paper (1.3cm wide) soaked in 25% FeCl_3_ saturated solution and a small piece of Para membrane (1.7 cm wide) was put under the artery for 15 min to induce the formation of carotid artery thrombosis. After blood reperfusion for one hour, NS solution (0.3 mL/kg), FITC-UK/PD-RGDS complex solution (8000 U/kg), or FITC-UK solution (8000 U/kg) were injected via the femoral vein. The rats were sacrificed 1-h post-administration. The liver, spleen, kidneys, lungs, heart, and thrombus clots are separated and taken out. About 1 g of each tissue was added into 3 mL homogenizing buffer (0.32 M sucrose, 100 mM HEPES, pH 7.4), and homogenized in a glass homogenizer. After homogenization, the liquid was centrifuged (4000 rpm for 15 min) to obtain the supernatant samples of each tissue. The supernatant powder samples of each tissue were obtained by freeze-drying. The appropriate amount of supernatant powder samples of each tissue was dissolved in an appropriate solution (1% Triton X-100, 100 mM NaCl, 0.1% SDS, 0.5% Na-Deoxycholate) and cultured at 4 degrees for 30 min to obtain the final test samples of organs.

Standard solutions of 20, 10, 5, 2.5, 1, 0.5, 0.25, and 0.1 mg/L FITC-UK were prepared with 10 mM PBS buffer solution (pH 7.4). The fluorescence intensity was measured at the excitation wavelength (493.0 nm) and emission wavelength (524.0 nm). The concentration of FITC-UK in each tissue was determined by converting measurements to concentrations through a standard curve between the fluorescence intensity versus the concentration of the UK.

The described assessments were approved by the Ethics Committee of Capital Medical University. The committee assures the welfare of the animals was maintained under the requirements of the Animal Welfare Act and according to the guideline for the care and use of laboratory animals.

## 5. Conclusions

RGD sequence peptides could have specific affinities to activate platelets in the blood clot. In the present study that the UK/PD-RGDS complex delivery system was constructed by PD-RGDS containing multiple RGDS based on the hypothesis of increasing the thrombolytic activity and decreasing the bleeding complications. Results showed that in vivo, the UK/PD-RGDS complex group greatly improved the thrombolytic activity compared with the UK group and significantly reduced the bleeding at the same time. Nanoscale agents could help to penetrate the thrombus much deeper and loosen the fibrin network in the thrombus [12]. The UK/PD-RGDS complex nanosphere (270–277 nm) could penetrate the thrombus deeper and have better thrombolytic efficiency. Secondly, the thrombolysis efficiency of UK/PD-RGDS complex carrying RGDS peptides was higher [15]. The cRGD liposomes could significantly reduce the dose of urokinase by 75% [8]. The UK/PD-RGDS complex could decrease the dose of urokinase by 90%, which may be because of the high grafting rate of PD-RGDS (46%). An assay of thrombolytic potency showed that complexation of UK and PD-RGDS improved the thrombolytic potency of UK. The results of the organ distribution study revealed that the FITC-UK/PD-RGDS complex group has more FITC-UK levels in thrombus than the UK group. The in vitro thrombolytic activity of the UK/PD-RGDS complex group was reserved. The UK/PD-RGDS complex improved the thrombolytic effect of thrombolytic agents and could decrease the dose of UK by 90%. Thus, the UK/PD-RGDS complex is a promisingly safe and effective targeting delivery method for the UK.

The UK/PD-RGDS complex as the target material could be feasible to carry UK and carry out the thrombolytic therapy because it reduced the side effects of bleeding and the risk of bleeding complications.

## Data Availability

Not applicable.

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
