# Peer review of "Poly-α, β-d, l-Aspartyl-Arg-Gly-Asp-Ser-Based Urokinase Nanoparticles for Thrombolysis Therapy"

_molecules, 2023, doi:10.3390/molecules28062578_

Round 1

Reviewer 1 Report

The manuscript “Poly-α, β-D, L-aspartyl-Arg-Gly-Asp-Ser-based Urokinase Nanoparticles for Thrombolysis Therapy” written by Yuji Wang, Ming Zhao, et al. in general is a solid work on the improvement of known thrombolytic agents by incorporating them in nanoparticle system to improve targeting and reducing side effects. However, the manuscript in its current state seems to be too raw for publication. The following questions should be answered before the final decision:
1.    How the size of nanoparticles was measured? Is the phrase “The particle sizes therein showed that the UK/PD-RGDS complex (1:1 w/w) had minimum fluctuation (272.0 ± 4.66 nm)” consistent with the DLS profiles (figure 1, E)?
2.    The following phrase is highly controversial: “Because UK/PD-RGDS complex group (2 000 IU/kg) had similar thrombolytic activity to the UK group (20 000 IU/kg) (P >0.05), the UK/PD-RGDS group complex greatly improved the thrombolytic activity compared with UK group.”
3. Where are the error bars for figure 5?
Also, it is highly recommended to improve the quality of pictures for any next submission. Please, indicate all statistical values in standard format: text… value 1 mean ± SD … text … value 2 mean ± SD (P value, samples N) everywhere in the text where the comparison is made.

Author Response

Dear reviewer 1,

Thank you for your comments on our manuscript entitled “Poly-α,β-D, L-aspartyl-Arg-Gly-Asp-Ser-based Urokinase Nanoparticles for Thrombolysis Therapy” (Manuscript Number: molecules-2253431). These comments are all valuable, helpful to revise and perfect our paper, and also have important guiding significance for our research. We have carefully studied these comments and made corrections to the best of our ability, which we hope will be approved. Modifications are marked using the "Track Changes" function in the paper. The main corrections in the paper and responses to your comments are as follows.

Reviewer 1: 

Comment 1: How the size of nanoparticles was measured? Is the phrase “The particle sizes therein showed that the UK/PD-RGDS complex (1:1 w/w) had minimum fluctuation (272.0 ± 4.66 nm)” consistent with the DLS profiles (Figure 1, E)?

Response: Thanks for your opinion!

The size of nanoparticles was measured as follows:

1) Prepare solution of UK/PD-RGDS complex at a concentration of 2.67 mg/mL;

2) Determine the z-coverage size of UK/PD-RGDS complex solution on a Malvern's Zeta Sizer (Nano-ZS90) by automatic measurement mode (n=3).

Minimum fluctuation” does not refer to the particle size distribution width of complex solution (Figure 1E). “Minimal fluctuation” means that the z-coverage size of the UK/PD-RGDS complex (1:1 w/w) changed less within 48 hours compared with the other two ratios of complex solutions. For example, the z-coverage size of complex solutions with ratios of 1:3 and 1:5 decreased significantly at 48 hours. But the z-average sizes of the UK/PD-RGDS complex solution varied between 281.7-266.5 nm within 48 hours at a ratio of 1:1 (Figure 1A). Therefore, we claimed that “the UK/PD-RGDS complex (1:1 w/w) had minimum fluctuation”.

To accurately describe the content, we made the following changes:

LOC 1: page 4, line 4-5, “The particle sizes there showed that the UK/PD-RGDS complex (1:1 w/w) had minimum fluctuation (272.0 ± 4.66 nm) among three UK/PD-RGDS complexes.” has been changed to “The z-coverage sizes therein showed that the UK/PD-RGDS complex (1:1 w/w) (272.0 ± 4.66 nm) has a smaller range of variation over 48 hours compared to the other two UK/PD-RGDS complexes ratio”.

LOC 2: page 4, the title of Figure 1E, "The particle size of the UK/PD-RGDS complex" has been changed to “The size distribution of the UK/PD-RGDS complex (1:1) was measured by dynamic light scattering method”.

LOC 3: pages 3-5, all "particle size" measured by dynamic light scattering method in the titles of Figure 1, A, F, table 1, and texts have been changed to “z-coverage size”.

Comment 2: The following phrase is highly controversial: “Because UK/PD-RGDS complex group (2 000 IU/kg) had similar thrombolytic activity to the UK group (20 000 IU/kg) (P >0.05), the UK/PD-RGDS group complex greatly improved the thrombolytic activity compared with UK group.”

Response: Thanks for your suggestion. To clearly describe this content, we made the following changes:

LOC 9: page 6-7, “The in vivo thrombolytic activity results (Figure 3C) showed that the UK group (20 000 IU/kg) had a significant thrombolytic activity due to the reduction of thrombolysis weight (29.3 ± 2.71 mg) compared with the NS group (P < 0.05), while UK/PD-RGDS complex group at a dose of 2 000 IU/kg showed significant thrombolytic activity (P < 0.05) according to the loss of thrombolysis weight (30.2 ± 3.00 mg). Because UK/PD-RGDS complex group (2 000 IU/kg) had similar thrombolytic activity to the UK group (20 000 IU/kg) (P >0.05), the UK/PD-RGDS group complex greatly improved the thrombolytic activity compared with UK group.” has been changed to: “Figure 3C showed the in vivo thrombolytic activity results of both positive control and our sample. Initially, we used a dosage of 20 000 IU/Kg (the clinical dose of UK) in the in vivo assay to compared the thrombosis activity between our sample (UK/PD-RGDS complex at 1:1 ratio) and the commercial UK. However, the result showed that both of positive control (UK) and our sample (UK/PD-RGDS complex) are active and have similar in vivo thrombolytic potency (Figure 3C). We thought that both UK and our sample may reach their maximal thrombolytic activity at 20 000 IU/kg. To assess whether our sample is more potent than commercial UK, we lowered the dose of our sample to 2 000 IU/kg and evaluated them in a second in vivo assay. Compared with NS group, there was no significant difference in thrombus weight reduction in UK (2 000 IU/kg) group (P>0.05). The results showed that low dose of our sample was as effective as high doses, whereas low dose of UK was inactive (Figure 3D). Therefore, we claimed that the thrombolytic activity of UK/PD-RGDS complex was enhanced compared to UK.”

Comment 3: Where are the error bars for figure 5? Also, it is highly recommended to improve the quality of pictures for any next submission. Please, indicate all statistical values in standard format: text… value 1 mean ± SD … text … value 2 mean ± SD (P value, samples N) everywhere in the text where the comparison is made.)

Response: “Where are the error bars for figure 5?” Thanks for pointing out the missing error bars in figure 5. The problem is caused by a display issue. Now we changed the color of the error bars to black and solved the problem (Figure 5).

LOC 4: pages 8-10, Figure 4, Figure 5 and Figure 6, we have improved their quality.

LOC 5: pages 1-4, we have revised “mean” to “mean ± SD”.

Again, we are very grateful to you for your excellent comments. We have tried our best to improve the manuscript. We sincerely thank you for your enthusiastic work, and hope that the revision will be approved.

Sincerely yours,

Yuji Wang, Ph. D., Full Professor

Dean of School of Pharmaceutical Sciences

Capital Medical University

E-mails: [email protected]

Reviewer 2 Report

Manuscript review "Poly-α, β-D, L-aspartyl-Arg-Gly-Asp-Ser-based Urokinase Nanoparticles for Thrombolysis Therapy" written by Shuangling Chen, Meng Liang, Chengli Wu, Xiaoyi Zhang, Yuji Wang, Ming Zhao . The manuscript is devoted to an important medical problem of our time - the fight against thrombosis. On the whole, the manuscript is written clearly and understandably, but I would like to draw the attention of the authors to a number of points that can be improved.
1. In the introduction part of the manuscript, there is no information about alternative developments in this field of knowledge. Thus, a number of polymers containing antithrombotic enzymes were previously developed. Moreover, these materials were capable of sustained release of enzymes (10.1080/09205063.2020.1760699 and 10.3390/ma1224107). I think that the authors should tell the readers what was done before them by other researchers.
2. Authors do not correctly present their results to readers. For example, PD-RGDS (278 ± 3.772 nm). How can there be a measurement to units of nanometers, and SEM is calculated with an accuracy of pm? This needs to be fixed!
3. There are two figures 1 in the text of the manuscript. It needs to be corrected!
4. The size of nanoparticles decreases by 10% in 2 days (Figure 1A). The authors need to somehow explain this or bring a “whisker” on the graph. Maybe there is no reduction in size, but we see only a measurement error.
5. In the TEM photograph, we see nanoparticles with a size of about 10 nm, but there is no such size in the DLS plot. How can this be explained?
6. The authors compare their results with heparin (designated as NS in the manuscript). What's the point of this? Perhaps you need to provide negative control data?
7. The authors should write more clearly how the data presented in fig. 3C and fig. 5. It's not easy to figure it out!

Author Response

Dear reviewer 2,

Thank you for your comments! Please see the attachment.

Sincerely yours,

Yuji Wang, Ph. D., Full Professor

Dean of School of Pharmaceutical Sciences

Capital Medical University

E-mails: [email protected]

Round 2

Reviewer 1 Report

I believe, almost all questions were answered (please, add the number of experiments for every P value you mentioned).

Reviewer 2 Report

I recommend to accept